



# Hyperpolarization and the Physical Boundary of Liouville Space

Malcolm H. Levitt[1] and Christian Bengs[1]

[1]School of Chemistry, University of Southampton, SO17 1BJ, UK

**Correspondence:** Malcolm H. Levitt (mhl@soton.ac.uk)

**Abstract.** The quantum state of a spin ensemble is described by a density operator, which corresponds to a point in the Liouville space of orthogonal spin operators. Valid density operators are confined to a particular region of Liouville space, which we call the physical region, and which is bounded by multidimensional figures called simplexes. Each vertex of a simplex corresponds to a pure-state density operator. We provide examples for spins $I = 1/2$, $I = 1$, $I = 3/2$, and for coupled pairs of spins-1/2. We use the von Neumann entropy as a criterion for hyperpolarization. It is shown that the inhomogeneous master equation for spin dynamics leads to non-physical results in some cases, a problem that may be avoided by using the Lindbladian master equation.



# 1 Introduction

The central object of interest in NMR theory is the density operator, which describes the quantum state of the ensemble of spin systems. It is defined as follows:

$$\rho = \overline{|\psi\rangle \langle\psi|} \tag{1}$$

Here $|\psi\rangle$ specifies the quantum state of each individual spin system, and the overbar indicates an ensemble average (Ernst et al. (1987)). When expressed as a matrix in the eigenbasis of the coherent spin Hamiltonian, the diagonal elements are the spin

state populations, and the off-diagonal elements are the coherences between the spin states.

It is often useful to express the density operator as a superposition of orthogonal spin operators. For example, the highly influential paper by Sørensen, Bodenhausen, Ernst and co-workers advocates an expansion in terms of Cartesian product operators (Sørensen et al. (1984); Ernst et al. (1987)), while some other groups favour spherical tensor operators (Sanctuary (1976, 1980); Sanctuary and Temme (1985a, b); Bowden and Hutchison (1986b); Bowden et al. (1986); Bowden and Hutchison

(1986a, 1987); Bowden et al. (1990); Bain (1978, 1980a, b); Philp and Kuchel (2005); Garon et al. (2015)). In all cases, the density operator is written in the form

$$\rho = \sum_{q=1}^{N_L} \rho_q Q_q \tag{2}$$

where the coefficients $\rho_q$ are complex numbers in general, and the basis operators $Q_q$ are orthogonal:

$$\left(Q_q \middle| Q_{q'}\right) = \mathrm{Tr}\left\{Q_q^\dagger Q_{q'}\right\} = \delta_{qq'} \|Q_q\|^2 \tag{3}$$

The Kronecker-delta symbol $\delta_{ab}$ takes the value 1 for $a = b$, and 0 otherwise. The norm of the operator $Q_q$ is defined as $\|Q_q\| = \mathrm{Tr}\left\{Q_q^\dagger Q_q\right\}^{1/2}$.

The coefficients $\rho_q$ are often given evocative names which suggest their physical interpretation, for example "antiphase order", "zz-order", "spin alignment", "Zeeman polarization", "singlet order", and so on.

Since such expansions are nearly universal in modern NMR theory, it seems natural to pose questions of the form: "what

values may the coefficients $\rho_q$ take"?; "are the values of $\rho_q$ unlimited, or bounded in some way"?; "does the value of one coefficient influence the possible values of a second coefficient?", etc. Surprisingly, these natural questions are rarely posed in the NMR world, although they have not escaped the attention of mathematical physicists and applied mathematicians (Byrd and Khaneja (2003); Kimura and Kossakowski (2005); Bengtsson and Zyczkowski (2006); Goyal et al. (2016); Szymański et al. (2018)).

The expansion in equation 2 identifies the density operator with a point in a multidimensional space with coordinates $\{\rho_1, \rho_2 \ldots\}$. This space has been called *Liouville space* (Jeener (1982); Ernst et al. (1987)). In this article, we show that valid density operators may only be identified with points in a defined region of Liouville space which we call the *physical region*. The physical region is enclosed by a convex boundary, which we call the *physical boundary of Liouville space*. We ask: what is the shape of the physical boundary? Does it have straight edges, or is it spherical in all dimensions? (Spoiler: at least some

edges are straight).



In addition, we express some views on the nature and definition of hyperpolarization. For example, is pure parahydrogen hyperpolarized, even though it generates no NMR signal? (Spoiler: our answer is yes).

## 2 Orthonormal operators

To facilitate the discussion, the basis operators $Q_q$ in equation 2 are henceforth considered to be normalized as well as orthogonal, so that equation 3 is replaced by the simpler form:

$$\left(Q_q \middle| Q_{q'}\right) = \mathrm{Tr}\left\{Q_q^\dagger Q_{q'}\right\} = \delta_{qq'} \tag{4}$$

Note, however, that the Cartesian product operators advocated by Sørensen et al. (1984) are *not* normalized.

In general, $N_L$ operators are required in the expansion of equation 2, where $N_L = N_H^2$ and $N_H$ is the dimension of the Hilbert space of the individual spin systems. The orthonormal operators $\{Q_1, Q_2 \dots Q_{N_L}\}$ define a $N_L$-dimensional Liouville space (Jeener (1982); Ernst et al. (1987)). The density operator may be represented as a point with coordinates $\{q_1, q_2 \dots q_{N_L}\}$ in this space. All spin dynamics may be represented as a trajectory traced by the spin density operator as it moves through this abstract space.

Brief consideration shows that there must be limits to the physical region of Liouville space. Consider for example an ensemble of isolated spins-1/2. In this case, the dimension of Hilbert space is $N_H = 2$, the dimension of Liouville space is $N_L = 4$. The following four normalized operators may be chosen as the basis of Liouville space:

$$Q_1 = 2^{-1/2}\mathbb{1} \qquad\qquad Q_2 = 2^{1/2}I_x$$
$$Q_3 = 2^{1/2}I_y \qquad\qquad Q_4 = 2^{1/2}I_z \tag{5}$$

Since $\mathrm{Tr}\{\rho\} = 1$ by definition, the first coefficient is fixed at $q_1 = 2^{-1/2}$. So the density operator is only free to move in the subspace formed by the other three operators, $\{Q_1, Q_2, Q_3\}$, which are proportional to the angular momentum operators in the three Cartesian directions.

The populations of the two Zeeman states are given by

$$\langle\alpha|\rho|\alpha\rangle = 2^{-1/2}(q_1 + q_4)$$
$$\langle\beta|\rho|\beta\rangle = 2^{-1/2}(q_1 - q_4) \tag{6}$$

Both state populations are, by definition, bounded by 0 and 1:

$$0 \leq \langle\alpha|\rho|\alpha\rangle \leq 1$$
$$0 \leq \langle\beta|\rho|\beta\rangle \leq 1 \tag{7}$$

Hence the coefficient $q_4$ is bounded as follows:

$$-2^{-1/2} \leq q_4 \leq 2^{-1/2} \tag{8}$$





The upper bound $q_4 = 2^{-1/2}$ corresponds to maximum spin polarization along the positive z-axis, with the $|\alpha\rangle$ state completely populated, and the $|\beta\rangle$ state completely depleted. The lower bound $q_4 = -2^{-1/2}$ corresponds to maximum spin polarization along the negative z-axis, with the $|\alpha\rangle$ state completely depleted, and the $|\beta\rangle$ state completely populated.

In the case of isolated spins-1/2, the physical bounds on Liouville space are therefore defined by the fixing of one coordinate ($q_1 = 2^{-1/2}$) and the constraint of the other three to the interior of a sphere of radius $2^{-1/2}$. Within a numerical factor, this geometrical bound is of course identical to the familiar Bloch sphere – the seminal geometrical object in magnetic resonance theory.

What about systems other than spins-1/2? Liouville space has more than three dimensions in such cases, and is hard to visualise. Nevertheless it is tempting to assume that the physical bounds are still spherical, albeit with an extension to higher dimensions. However, this turns out to be incorrect, in general. The physical bounds for the zero-quantum parts of the density operator turn out not to be spheres but *regular simplexes*. A regular simplex in one dimension is a line, a regular simplex in two dimensions is an equilateral triangle, a regular simplex in three dimensions is a regular tetrahedron, with the concept extending to arbitrary dimensions. In general, a simplex is the simplest possible convex object, where the term *convex* means that any two points belonging to the object may be connected by a straight line which never leaves the object. In general, a simplex in $N$ dimensions is called a $N$-simplex, although some simplexes also have special names, such as the line (1-simplex), the triangle (2-simplex), the tetrahedron (3-simplex), and the *pentachoron* or *5-cell* (4-simplex) (Coxeter (1963)).

The physical boundary of Liouville space is of little consequence for "conventional" NMR experiments, which are performed at or near thermal equilibrium. This is a region exceedingly close to the origin of Liouville space (except for the fixed projection onto the unity operator) and hence very far from the boundary. However, hyperpolarization techniques such as optical pumping (Kastler (1957); Navon et al. (1996)), dynamic nuclear polarization (Griffin and Prisner (2010); Ardenkjaer-Larsen et al. (2003); Jannin et al. (2012)), quantum-rotor-induced polarization (Icker and Berger (2012); Meier et al. (2013); Dumez et al. (2015)) and parahydrogen-induced polarization (Bowers and Weitekamp (1987); Adams et al. (2009)), have provided ready access to regions which are "close to the edge". Furthermore, spin systems which are in a highly non-equilibrium state are of great practical importance, because of the greatly enhanced NMR signals that they can be produce. The position and shape of the physical boundary has therefore become relevant.

Furthermore, some familiar concepts in magnetic resonance which were originally developed in the context of near-equilibrium spin dynamics do not retain validity far from equilibrium. An important case is the inhomogeneous master equation (Ernst et al. (1987)), which fails close to the physical boundary of Liouville space, where it should be replaced by a Lindbladian master equation (Bengs and Levitt (2020)).



## 3 Polarization Moments

### 3.1 Isolated spins-$I$

For an ensemble of isolated spins-$I$, a suitable expansion of the form in equation 2 is as follows:

$$\rho = \sum_{\lambda=0}^{2I} \sum_{\mu=-I}^{+I} \rho_{\lambda\mu} \mathbb{T}_{\lambda\mu} \tag{9}$$

Here $\rho_{\lambda\mu}$ are complex numbers which are called here *polarization moments*, following the usage in the atomic physics community (Budker et al. (2002); Auzinsh et al. (2014)). The operators $\mathbb{T}_{\lambda\mu}$ are *normalized irreducible spherical tensor operators* (NISTOs). They are normalized over the spin-$I$ Hilbert space:

$$\left(\mathbb{T}_{\lambda\mu}\middle|\mathbb{T}_{\lambda\mu}\right) = \mathrm{Tr}\{\mathbb{T}_{\lambda\mu}^{\dagger}\mathbb{T}_{\lambda\mu}\} = 1 \tag{10}$$

The normalized spherical tensor operators $\mathbb{T}_{\lambda\mu}$ differ from the operators $T_{\lambda\mu}$ commonly used in NMR theory (Spiess (1978); Mehring (1976)) by a multiplicative factor.

For isolated spins-$I$, the low-rank normalized spherical tensor operators are as follows, for the case $\mu = 0$:

$$\mathbb{T}_{00} = \mathbb{N}_0^I \mathbb{1}$$
$$\mathbb{T}_{10} = \mathbb{N}_1^I I_z$$
$$\mathbb{T}_{20} = \mathbb{N}_2^I \frac{1}{\sqrt{6}} \left(3I_z^2 - I(I+1)\mathbb{1}\right)$$
$$\mathbb{T}_{30} = \mathbb{N}_3^I \frac{1}{\sqrt{10}} \left(5I_z^3 + (1 - 3I(I+1))I_z\right) \tag{11}$$

The normalization factors are as follows:

$$\mathbb{N}_0^I = (2I+1)^{-1/2}$$
$$\mathbb{N}_1^I = \{\frac{1}{3}I(I+1)(2I+1)\}^{-1/2}$$
$$\mathbb{N}_2^I = \{\frac{1}{30}I(I+1)(2I-1)(2I+3)(2I+1)\}^{-1/2}$$
$$\mathbb{N}_3^I = \{\frac{1}{70}I(I+1)(I-1)(I+2)(2I-1)(2I+3)(2I+1)\}^{-1/2} \tag{12}$$

These normalization factors depend on the spin quantum number $I$ and the rank $\lambda$, but are independent of the component index $\mu$.

It follows from equations 2 and 10, and the orthogonality of the NISTOs, that any polarization moment may be derived from the density operator by a Liouville bracket operation:

$$\rho_{\lambda\mu} = \left(\mathbb{T}_{\lambda\mu}\middle|\rho\right) = \mathrm{Tr}\{\mathbb{T}_{\lambda\mu}^{\dagger}\rho\} \tag{13}$$

The polarization moments have the following symmetry:

$$\rho_{\lambda\mu} = (-1)^{-\mu}\rho_{\lambda-\mu}^* \tag{14}$$



which follows from the hermiticity of the density operator and the symmetries of the spherical tensor operator components (Varshalovich et al. (1988)).

For isolated spins-$I$, the condition $\mathrm{Tr}\{\rho\} = 1$ fixes the value of the rank-0 polarization moment:

$$\rho_{00} = (2I+1)^{-1/2} \tag{15}$$

120 The rank-1 polarization moment $\rho_{10}$ is proportional to the z-polarization of the spin-$I$ ensemble as follows:

$$\rho_{10} = \left\{\frac{1}{3I}(I+1)(2I+1)\right\}^{-1/2} p_z \tag{16}$$

Similarly, the rank-1 polarization moments $\rho_{1\pm1}$ are proportional to complex combinations of the transverse polarizations:

$$\rho_{1\pm1} = \left\{\frac{2}{3I}(I+1)(2I+1)\right\}^{-1/2} (\mp p_x + i p_y) \tag{17}$$

The relationship in equation 16 evaluates as follows for some common spin quantum numbers $I$:

$$
\begin{aligned}
\rho_{10} &= 2^{-1/2}\, p_z & (I = 1/2)\\
\rho_{10} &= 2^{-1/2}\, p_z & (I = 1)\\
\rho_{10} &= \frac{3}{2}5^{-1/2}\, p_z & (I = 3/2)\\
\rho_{10} &= \left(\frac{5}{14}\right)^{1/2} p_z & (I = 5/2)
\end{aligned} \tag{18}
$$

125

In atomic physics, finite moments with rank $\lambda = 1$ are called *orientation*, while finite moments with rank $\lambda = 2$ are called *alignment* (Auzinsh et al. (2014)). Although the term *orientation* is not generally used for this purpose in the magnetic resonance community, the term *alignment* is used to imply rank-2 multipole order, particularly in the context of solid-state NMR as applied to quadrupolar nuclei (Batchelder (2007)). For isolated spins-$I$, the terms *spin alignment* and *quadrupolar order* may

130 be regarded as synonymous.

Multipole expansions of the spin density operator as in equation 2 have long been used in NMR. Extensive theoretical development was performed by Sanctuary, Bowden, Bain and co-workers (Sanctuary (1976, 1980); Sanctuary and Temme (1985a, b); Bowden and Hutchison (1986b); Bowden et al. (1986); Bowden and Hutchison (1986a); Bowden et al. (1990); Bain (1978, 1980a, b)), and has been exploited to generate graphical representations of density operator evolution (Philp

135 and Kuchel (2005); Garon et al. (2015)). One of the salient early examples of the multipole description is the treatment of quadrupolar relaxation by Bodenhausen and co-workers (Jaccard et al. (1986)). In this elegant paper, the relaxation dynamics of quadrupolar nuclei outside the extreme narrowing limit is treated in terms of propagation in the space of spherical tensor operators, drawing fruitful parallels with the concepts of coherence transfer pathways (Bodenhausen et al. (1984); Bain (1984)).

There are also techniques for determining the polarization moments of a spin ensemble *experimentally* at any point during a

140 pulse sequence, by combining the signals from many successive experiments, multiplied by complex factors. This method has been called *spherical tensor analysis* (van Beek et al. (2005)) and has been applied to the study of endofullerenes (Carravetta et al. (2007)).





## 3.2  Spin-1/2 pairs

The construction of spherical tensor operators for systems of coupled spins is a complicated affair. Extensive expositions
of the technique have been given (Sanctuary (1976, 1980); Sanctuary and Temme (1985a, b); Bowden et al. (1990); Garon
et al. (2015)). In this article, the discussion of coupled spin systems is restricted to the simplest case, namely pairs of coupled
spins-1/2. Since the dimension of Hilbert space is $N_H = 4$, the dimension of Liouville space is $N_L = 16$. This space includes
6 orthogonal zero-quantum operators, four of which are symmetric with respect to spin exchange, and two of which are
antisymmetric. The four symmetric $\mu = 0$ operators are as follows:

$$\mathbb{T}_{00}^0 = \tfrac{1}{2}\mathbb{1}$$
$$\mathbb{T}_{00}^{12} = -2 \times 3^{-1/2}\,\mathbf{I}_1 \cdot \mathbf{I}_2$$
$$\mathbb{T}_{10}^+ = 2^{-1/2}(I_{1z} + I_{2z})$$
$$\mathbb{T}_{20}^{12} = (2/3)^{1/2}\big(3I_{1z}I_{2z} - \mathbf{I}_1 \cdot \mathbf{I}_2\big) \tag{19}$$

Note that spin-1/2 pairs support two different spherical tensor operators with rank $\lambda = 0$, denoted $\mathbb{T}_{00}^0$ and $\mathbb{T}_{00}^{12}$. The plus
superscript in $\mathbb{T}_{10}^+$ indicates that the operators $I_{1z}$ and $I_{2z}$ are combined with the same sign.

The operator $\mathbb{T}_{00}^0$ is proportional to the unity operator. The corresponding polarization moment is fixed by the condition
$\mathrm{Tr}\{\rho\} = 1$ to the value

$$\rho_{00}^0 = \tfrac{1}{2} \tag{20}$$

The operator $\mathbb{T}_{00}^{12}$ is proportional to the scalar product of the two spin angular momenta. The corresponding polarization
moment is given by

$$\rho_{00}^{12} = \frac{\sqrt{3}}{2}\,p_S \tag{21}$$

where $p_S$ is called the *singlet polarization* and corresponds to the difference between the population of the singlet state and
the mean population of the triplet manifold, for the spin-pair ensemble. In many cases, the singlet polarization is protected
against common relaxation mechanisms, and exhibits an extended lifetime (Carravetta et al. (2004); Carravetta and Levitt
(2004); Cavadini et al. (2005); Sarkar et al. (2007b, a); Ahuja et al. (2009); Levitt (2019); Dumez (2019)). The operator
$\mathbb{T}_{10}^+$ corresponds to a symmetric combination of the z-angular momentum operators for the two spins. The corresponding
polarization moment is proportional to the mean z-polarization of the spin ensemble:

$$\rho_{10}^+ = \frac{1}{\sqrt{2}}p_z \tag{22}$$

The operator $\mathbb{T}_{20}^{12}$ corresponds to the rank-2 spherical tensor operator of the coupled spin pair. The corresponding polarization
moment $\rho_{20}^{12}$ is proportional to the rank-2 order (dipolar order) of the spin-pair ensemble.



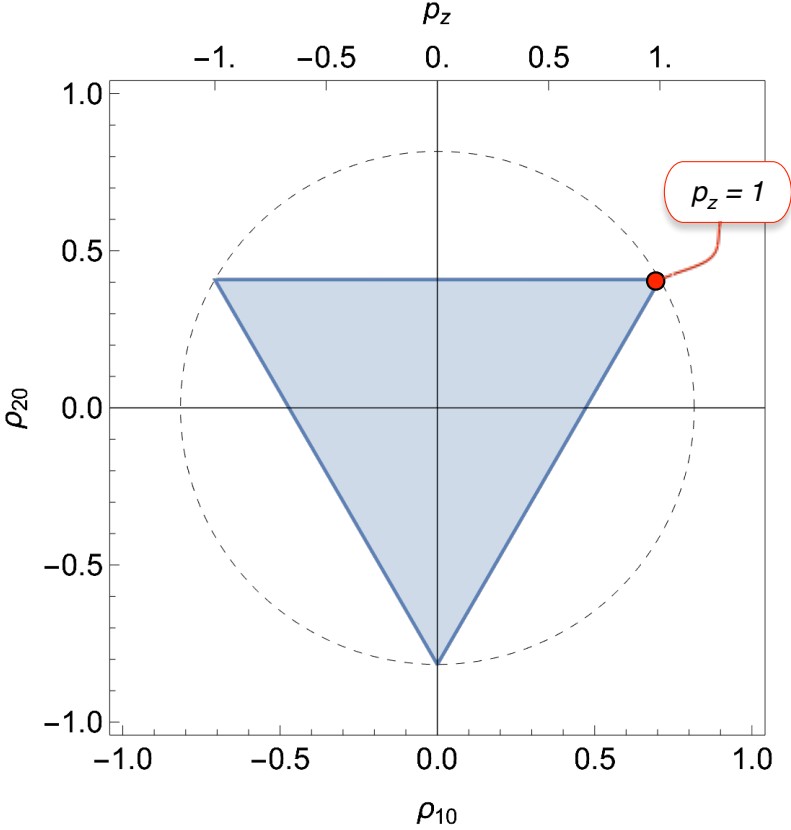

**Figure 1.** Physical bounds on the rank-1 and rank-2 polarization moments for isolated spins $I = 1$. The shaded triangle shows the physically accessible region. The vertices correspond to pure-state density operators for each of the three spin-1 Zeeman states. The radius of the dashed circle is $\sqrt{2/3}$. The rank-1 polarization moment $\rho_{10}$ is related to the z-polarization $p_z$ through equation 18. The red circle indicates a state with maximum Zeeman polarization.

## 4 Physical Bounds of Liouville Space

The population of each individual spin state is bounded by $0$ and $1$. For isolated spins-$I$, this leads to the following set of $2I$
170    simultaneous inequalities on the $\mu = 0$ polarization moments:

$$0 \leq \sum_{\lambda=1}^{2I} \langle I, M_I | \mathbb{T}_{\lambda 0} | I, M_I \rangle \rho_{\lambda 0} \leq 1 \qquad (23)$$

for $M_I \in \{+I, +I - 1 \ldots - I\}$. The system of inequalities in equation 23 defines the physical bounds of the $\mu = 0$ polarization moments.

     The consequences are now explored for some common spin systems.



### 4.1 Spins-1/2

For isolated spins-1/2, the rank-0 polarization moment is given from equation 14 by

$$\rho_{00} = 2^{-1/2} \qquad \text{for } I = 1/2 \tag{24}$$

Equation 23 leads to the following physical bounds for the rank-1 polarization moment:

$$-2^{-1/2} \leq \rho_{10} \leq +2^{-1/2} \qquad \text{for } I = 1/2 \tag{25}$$

From equation 16, this corresponds to the expected bounds on the z-polarization of the spin ensemble:

$$-1 \leq p_z \leq +1 \tag{26}$$

which should come as no surprise. No spin system may have more than 100% polarization.

### 4.2 Spins-1

The bounds on the polarization moments are more complicated for an ensemble of isolated spins-1. The rank-0 polarization moment is given through equation 14 by:

$$\rho_{00} = 3^{-1/2} \qquad \text{for } I = 1 \tag{27}$$

The inequalities on the rank-1 and rank-2 polarization moments evaluate as follows:

$$
\begin{aligned}
0 &\leq \frac{1}{3} + \frac{1}{\sqrt{2}}\rho_{10} + \frac{1}{\sqrt{6}}\rho_{20} \leq 1 \\
0 &\leq \frac{1}{3} - \sqrt{\frac{2}{3}}\rho_{20} \leq 1 \\
0 &\leq \frac{1}{3} - \frac{1}{\sqrt{2}}\rho_{10} + \frac{1}{\sqrt{6}}\rho_{20} \leq 1 \qquad \text{for } I = 1
\end{aligned}
\tag{28}
$$

The values of $\{\rho_{10}, \rho_{20}\}$ which satisfy the inequalities in equation 28 lie within the shaded triangle in figure 1. The vertices of the triangle have coordinates $\{\rho_{10}, \rho_{20}\}$ given by $\{\pm 2^{-1/2}, 6^{-1/2}\}$ and $\{0, -(2/3)^{1/2}\}$; Each vertex corresponds to a pure-state density operator, in which only one state is populated. Similar triangular bounds have been identified in the mathematics literature (Kimura and Kossakowski (2005); Goyal et al. (2016)).

The equilateral triangle in figure 1 corresponds to a *regular simplex* in two dimensions (Coxeter (1963)).

The maximum z-polarization of $p_z = 1$ corresponds to the upper-right vertex. Figure 1 shows that this highly-polarized state corresponds to a mixture of rank-1 polarization (Zeeman order) and rank-2 polarization (quadrupolar order). It follows that the near-complete hyperpolarization of spin-1 nuclei, as performed by the Bodenhausen group (Aghelnejad et al. (2020)) generates hyperpolarized quadrupolar order, as well as Zeeman order.



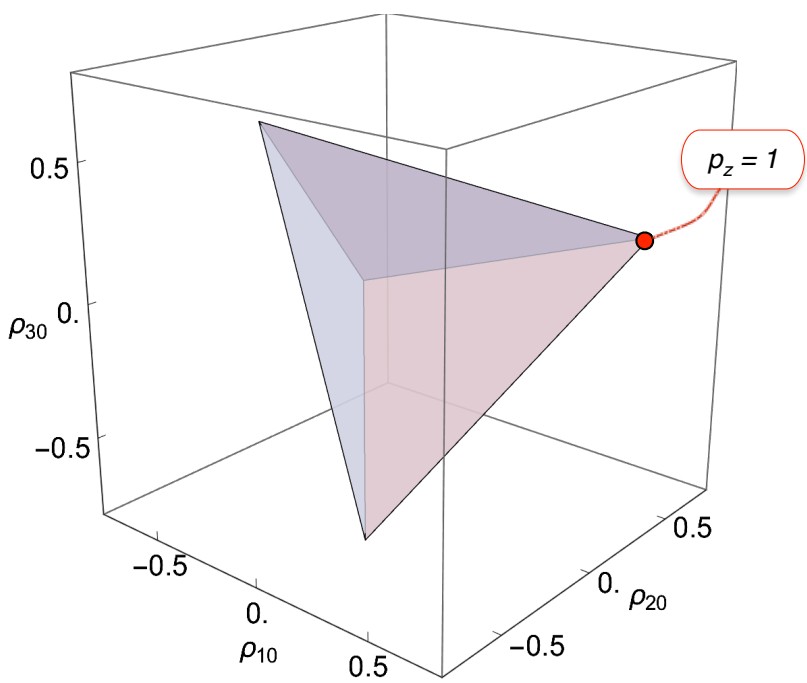

**Figure 2.** Physical bounds on the rank-1, rank-2, rank-3 polarization moments for isolated spins $I = 3/2$. The shaded tetrahedron shows the physically accessible region. The vertices correspond to pure-state density operators for each of the four spin-1 Zeeman states. The red circle indicates the position of maximum Zeeman polarization.

### 4.3 Spins-3/2

In the case of isolated spins-3/2, the rank-0 polarization moment is given by:

$$\rho_{00} = 4^{-1/2} = 1/2 \qquad \text{for } I = 3/2 \tag{29}$$

For spins-3/2, there may be a finite rank-3 polarization moment $\rho_{30}$ as well as the rank-1 and rank-2 terms. The physical bounds on these polarization moments are set by the following inequalities:

$$
\begin{aligned}
0 &\leq \frac{1}{20}\left(5 + 6\sqrt{5}\rho_{10} + 10\rho_{20} + 2\sqrt{5}\rho_{30}\right) \leq 1 \\
0 &\leq \frac{1}{20}\left(5 + 2\sqrt{5}\rho_{10} - 10\rho_{20} - 6\sqrt{5}\rho_{30}\right) \leq 1 \\
0 &\leq \frac{1}{20}\left(5 - 2\sqrt{5}\rho_{10} - 10\rho_{20} + 6\sqrt{5}\rho_{30}\right) \leq 1 \\
0 &\leq \frac{1}{20}\left(5 - 6\sqrt{5}\rho_{10} + 10\rho_{20} - 2\sqrt{5}\rho_{30}\right) \leq 1 \qquad \text{for } I = 3/2
\end{aligned}
\tag{30}
$$

The physical bounds on the three polarization moments constrain the spin density operator to the interior of the regular tetrahedron shown in figure 2. The vertices of the tetrahedron are at coordinates $\{\rho_{10}, \rho_{20}, \rho_{30}\}$ given by $\{\pm 1/2\sqrt{5}, -1/2, \mp 3/2\sqrt{5}\}$





and $\{\pm 3/2\sqrt{5}, 1/2, \pm 1/2\sqrt{5}\}$. Each vertex corresponds to a pure-state density operator, in which only one state is populated. The z-polarization is related to the rank-1 polarization moment $\rho_{10}$ through equation 18.

The tetrahedron in figure 2 corresponds to a *regular simplex* in three dimensions (Coxeter (1963)).

### 4.4 Higher spins

210 The treatment above is readily extended to higher spins. For isolated spins-$I$, the bounding figure is given by a *regular simplex* in $2I$ dimensions. For example, the 4-dimensional bounding simplex of the polarization moments for spin $I = 2$ is called a *5-cell* or *pentachoron*; The 5-dimensional bounding simplex of the polarization moments for spin $I = 5/2$ is called a *5-simplex* or *hexateron*, and so on. Regular high-dimensional polytopes have been exploited before in NMR, albeit in a different context (Mamone et al. (2010); Levitt (2010)).

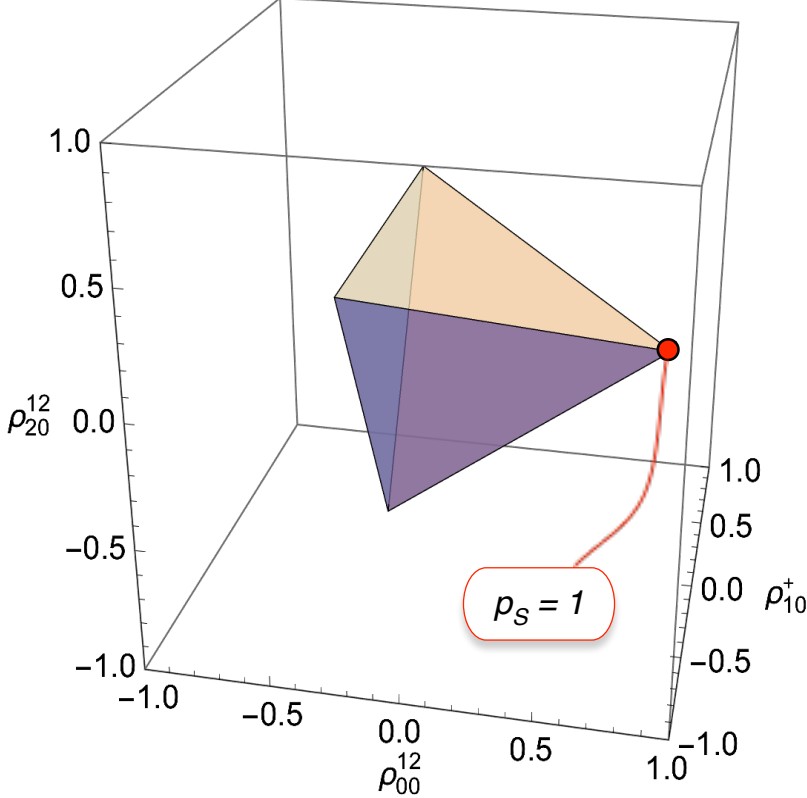

**Figure 3.** Physical bounds on the rank-0 polarization moment $\rho_{00}^{12}$, the rank-1 polarization moment $\rho_{10}^{+}$, and the rank-2 polarization moment $\rho_{20}^{12}$ for spin-1/2 pairs. The shaded tetrahedron shows the physically accessible region. The vertices correspond to pure-state density operators for each of the four singlet or triplet states. The annotation shows the vertex for exclusive population of the singlet state (corresponding to pure parahydrogen in the case of $H_2$ gas).



### 4.5 Spin-1/2 pairs

For spin-1/2 pairs, the symmetric $\mu = 0$ subspace is of dimension 4, spanned by the four symmetric spherical tensor operators given in equation 19. Since the polarization moment $\rho_{00}^0$ is fixed (equation 20), the symmetric part of the spin density operator may be described as a point with coordinates $\{\rho_{00}^{12}, \rho_{10}^+, \rho_{20}^{12}\}$, given by its projections onto the three orthonormal spherical tensor operators $\{\mathbb{T}_{00}^{12}, \mathbb{T}_{10}^+, \mathbb{T}_{20}^{12}\}$. The density operator may also include components that are antisymmetric with respect to exchange: these components lie outside this three-dimensional subspace, and are not considered further here.

The physical bounds on the symmetrical polarization moments $\{\rho_{00}^{12}, \rho_{10}^+, \rho_{20}^{12}\}$ are set by the following inequalities:

$$0 \leq \frac{1}{4}\left(1 + 2\sqrt{3}\rho_{00}^{12}\right) \leq 1$$

$$0 \leq \frac{1}{12}\left(3 - 2\sqrt{3}\rho_{00}^{12} + 6\sqrt{2}\rho_{10}^+ + 2\sqrt{6}\rho_{20}^{12}\right) \leq 1$$

$$0 \leq \frac{1}{12}\left(3 - 2\sqrt{3}\rho_{00}^{12} - 4\sqrt{6}\rho_{20}^{12}\right) \leq 1$$

$$0 \leq \frac{1}{12}\left(3 - 2\sqrt{3}\rho_{00}^{12} - 6\sqrt{2}\rho_{10}^+ + 2\sqrt{6}\rho_{20}^{12}\right) \leq 1 \qquad \text{for spin-1/2 pairs} \tag{31}$$

These inequalities constrain the polarization moments to the interior of the regular tetrahedron shown in figure 3. The vertices of the tetrahedron are at coordinates $\{\rho_{00}^{12}, \rho_{10}^+, \rho_{20}^{12}\}$ given by

$$\{\rho_{00}^{12}, \rho_{10}^+, \rho_{20}^{12}\} = \begin{cases} \{\frac{1}{2}3^{1/2}, 0, 0\} \\ \{-\frac{1}{2}3^{-1/2}, 2^{-1/2}, 6^{-1/2}\} \\ \{-\frac{1}{2}3^{-1/2}, -2^{-1/2}, 6^{-1/2}\} \\ \{-\frac{1}{2}3^{-1/2}, 0, -(2/3)^{1/2}\} \end{cases} \tag{32}$$

Each vertex corresponds to a pure-state density operator, in which the singlet state, or one of the three triplet states, is exclusively populated.

The highlighted point in figure 3 has coordinates $\{\frac{1}{2}3^{1/2}, 0, 0\}$. From equation 21, this point corresponds to unit singlet polarization ($p_S = 1$) and hence a pure singlet density operator:

$$\rho = |S_0\rangle \langle S_0| \tag{33}$$

where the singlet state is given by (Levitt (2019))

$$|S_0\rangle = \frac{1}{\sqrt{2}}\left(|\alpha\beta\rangle - |\beta\alpha\rangle\right) \tag{34}$$

A projection of the tetrahedral bound in figure 3 onto the $\{\rho_{00}^{12}, \rho_{10}^+\}$ plane is shown in figure 4. The corresponding values of the singlet polarization $p_S$ and z-polarization $p_z$ are shown on the axes, with the conversion factors given in equations 21 and 22. The red point in figure 4 shows that maximal singlet polarization is necessarily accompanied by zero z-polarization. In the case that the spin-1/2 pair is composed of the two proton nuclei of $H_2$, the red point corresponds to the spin density operator of pure parahydrogen (Farkas (1935)).

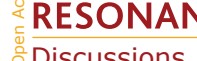

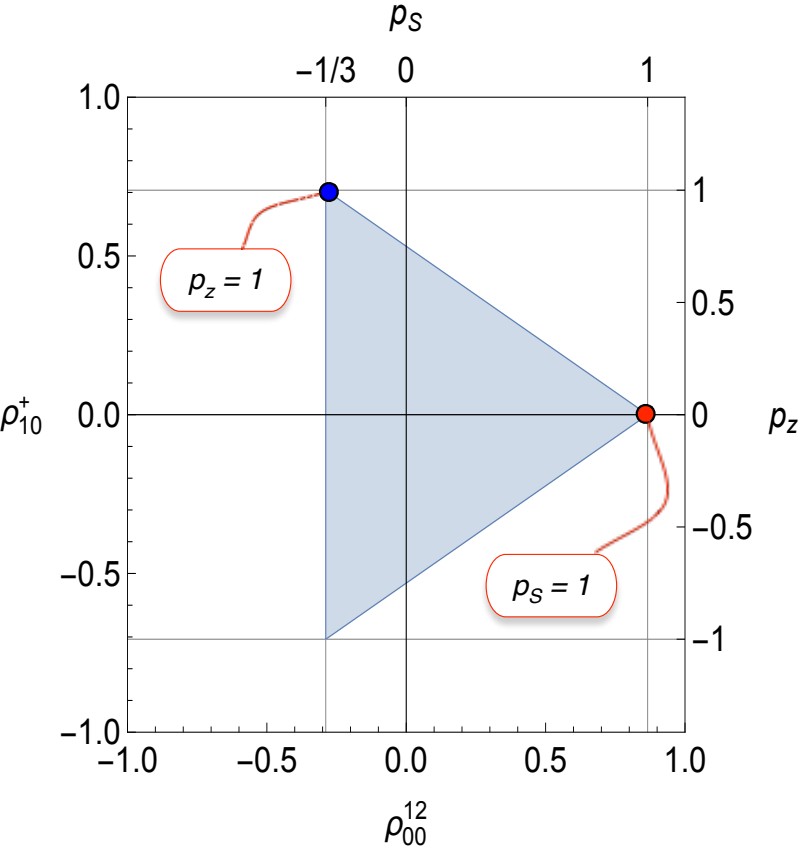

**Figure 4.** Physical bounds on the rank-0 polarization moment $\rho_{00}^{12}$ and the rank-1 polarization moment $\rho_{10}^+$ for spin-1/2 pairs. The corresponding values of the z-polarization $p_z$ and singlet polarization $p_S$ are also shown. The shaded triangle shows the physically accessible region. The annotations shows the vertices for exclusive population of the singlet state (red) and for maximal z-polarization (blue). Note that complete z-polarization is accompanied by singlet polarization of $p_S = -1/3$.

The blue point in figure 4 shows that maximal z-polarization is necessarily accompanied by singlet polarization of $p_S = -1/3$. This reflects the fact that maximal z-polarization can only be achieved by depleting the singlet state at the expense of
one of the triplet states. This fact may be exploited experimentally to generate hyperpolarized (negative) long-lived singlet order, by the application of low-temperature dynamic nuclear polarization to spin-pair systems (Tayler et al. (2012); Bornet et al. (2014); Mammoli et al. (2015)). Analogous phenomena are observed in more complex spin systems, such as methyl groups (Dumez et al. (2017)) and deuterated moieties (Kress et al. (2019)).



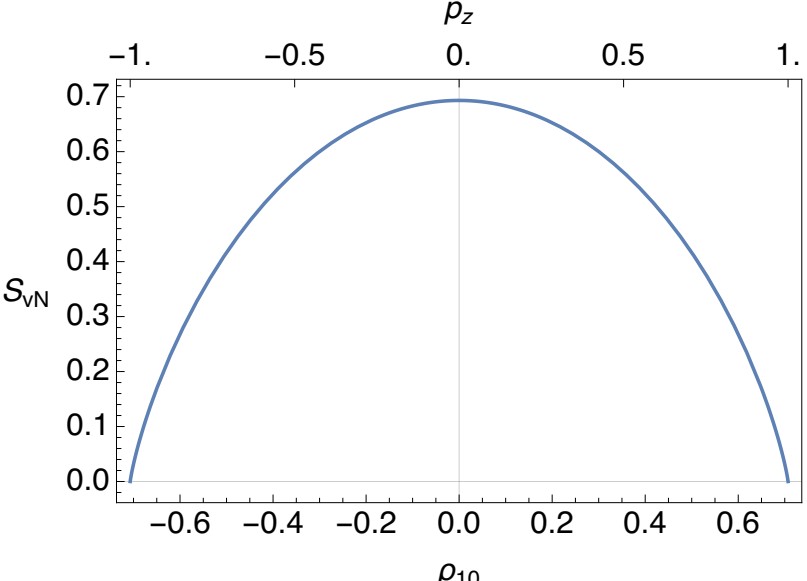

**Figure 5.** von Neumann entropy $S_{vN}$ plotted against the rank-1 polarization moment $\rho_{10}$ and the z-polarization $p_z$ for isolated spins-1/2, for the case of zero transverse polarizations $p_x = p_y = 0$. The maximum value of $S_{vN}$ is $\ln 2 \simeq 0.693$.

## 5   von Neumann Entropy

Quantum statistical mechanics uses the *von Neumann entropy* (vNE) to describe the disorder in, or absence of information about, a quantum system (Breuer and Petruccione (2010); Rodin et al. (2020)). It is derived from the spin density operator as follows:

$$S_{vN} = -\mathrm{Tr}\{\rho \ln \rho\} \tag{35}$$

The vNE for a system in a pure quantum state is zero, while the vNE for a system with equal populations of $N_H$ quantum
states, and no coherences, is given by $S_{vN} = \ln N_H$.

### 5.1   Spins-1/2

The von Neummann entropy $S_{vN}$ is plotted against the rank-1 polarization moment $\rho_{10}$ in figure 5, assuming that $\rho_{1\mu} = 0$ for $\mu = \pm 1$. The corresponding value of the z-polarization $p_z = \sqrt{2}\rho_{10}$ is shown on the top margin of the plot. The entropy goes to zero for complete z-polarization in the positive or negative sense ($p_z = \pm 1$), and attains the maximum value of $S_{vN} = \ln 2$
for zero polarization. The maximum entropy of $\ln 2$ reflects the equal populations of the two Zeeman eigenstates, and absence of coherences, for a completely saturated system.





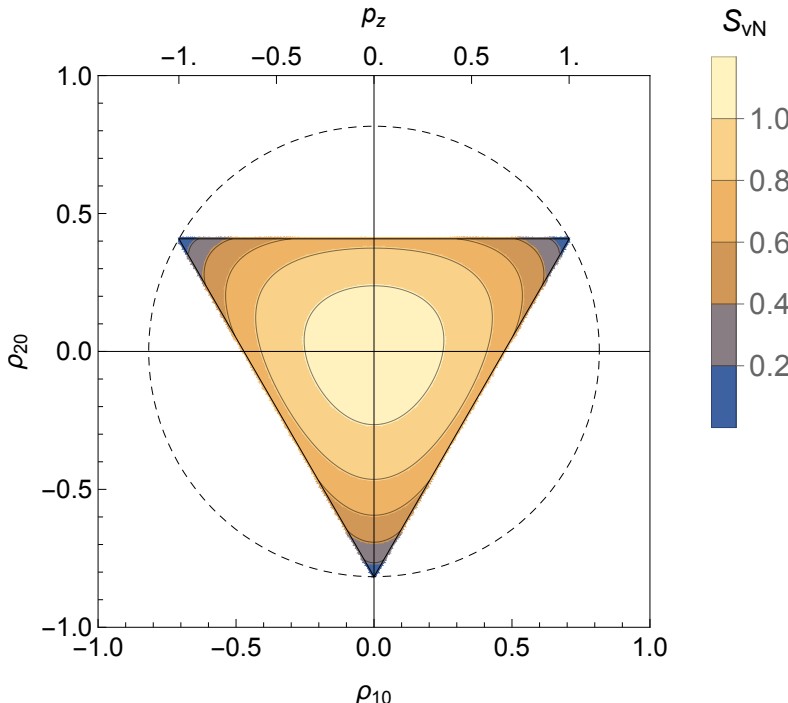

**Figure 6.** The von Neumann entropy $S_{vN}$ is shown as a contour plot against the rank-1 and rank-2 polarization moments for isolated spins-1, for the case of $\rho_{\lambda\mu} = 0$ for $\mu \neq 0$. Only the physical region is shown (see figure 1). The corresponding value of the z-polarization $p_z = 2^{1/2}\rho_{10}$ is shown along the top edge. The maximum value of the von Neumann entropy, reached at the origin, is $\ln 3 \simeq 1.10$.

### 5.2 Spins-1

For isolated spins-1, the von Neumann entropy is a function of the rank-1 and rank-2 polarization moments, assuming that all polarization moments $\rho_{\lambda\mu}$ vanish for $\mu \neq 0$. Figure 6 shows a contour plot of the von Neumann entropy against $\rho_{10}$ and

$\rho_{20}$, assuming that all polarization moments with $\mu \neq 0$ vanish. Only the physically allowed region is shown, delineated by the triangle, as in figure 1. The entropy goes to zero at the three vertices, which correspond to the pure-state density operators with 100% population of a single state. The von Neumann entropy reaches the maximum value of $\ln 3$ at the centre of the plot, corresponding to $\rho_{10} = \rho_{20} = 0$. The value of $\ln 3$ reflects the equal distribution of population over the three spin states.

### 5.3 Higher spins

The behaviour of the von Neumann entropy is readily anticipated for higher spin quantum numbers. The entropy vanishes at the $(2I+1)$ vertices of the $2I$-simplex which bounds the physical region. The entropy maximum of $\ln(2I+1)$ is reached at the origin of the space, which corresponds to equal populations for all of the $2I+1$ spin states.



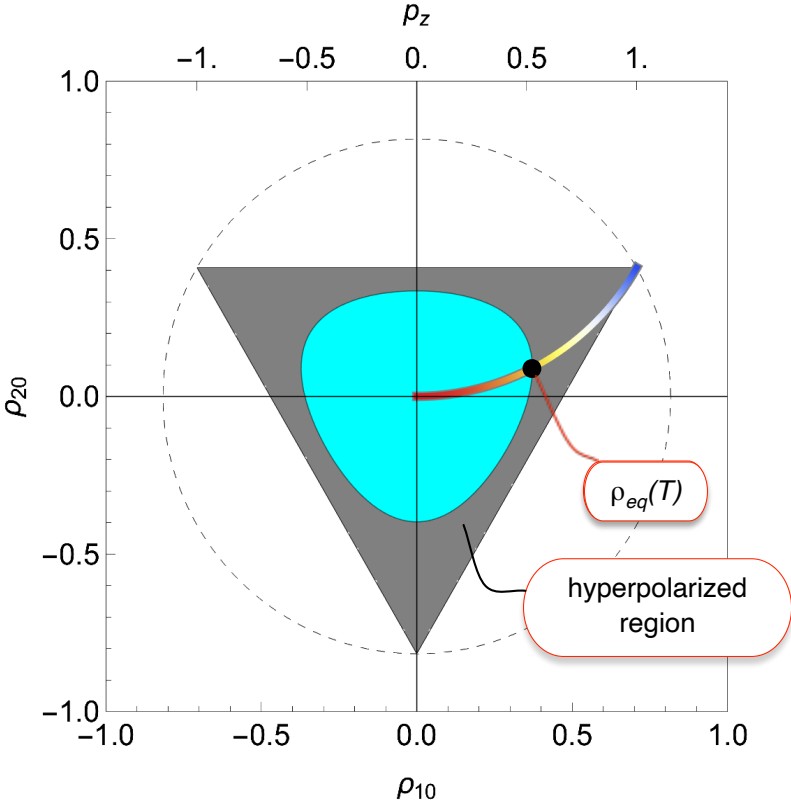

**Figure 7.** The coloured arc shows the set of thermal equilibrium density operators for spins-1 subjected to a dominant magnetic field. The spin temperature is indicated by colour, progressing from high (red) to low (blue). The black dot indicates the thermal equilibrium density operator at a particular temperature $T$. All points within the grey region represent hyperpolarized states of the spin-1 ensemble at temperature $T$.

## 6 Hyperpolarization

### 6.1 Thermal Equilibrium

Thermal equilibrium with the environment at temperature $T$ is reached when the density operator adopts the following form:

$$\rho_{\text{eq}}(T) = \frac{\exp\{-\hbar H_{\text{coh}}/k_B T\}}{\text{Tr}\{\exp\{-\hbar H_{\text{coh}}/k_B T\}\}} \tag{36}$$

where $H_{\text{coh}}$ is the coherent part of the spin Hamiltonian (excluding all fluctuating terms which drive dissipation). Equation 36 describes a Boltzmann distribution of spin-state populations under the coherent Hamiltonian $H_{\text{coh}}$.

In many cases, the coherent Hamiltonian is dominated by the Zeeman interaction with the main magnetic field, $H_{\text{coh}} \simeq \omega^0 I_z$,

where the Larmor frequency is $\omega^0 = -\gamma B^0$, and $B^0$ is the magnetic field. In this case the thermal equilibrium density operator



is given by

$$\rho_{\text{eq}} \simeq \frac{\exp\{\beta I_z\}}{\text{Tr}\{\exp\{\beta I_z\}\}} \tag{37}$$

where the normalized inverse temperature is $\beta = -\hbar\omega^0/k_B T$. Since equation 37 is non-linear in $I_z$, the thermal equilibrium density operator contains high-rank polarization moments in thermal equilibrium.

The coloured arc in figure 7 shows the set of thermal equilibrium density operators for a dominant Zeeman interaction (equation 37), over a range of spin temperatures. Blue denotes a low spin temperature ($\beta \to \infty$), while red denotes a high spin temperature ($\beta \to 0$). Note the increase of the rank-2 polarization moment $\rho_{20}$ at low spin temperatures.

## 6.2    A criterion of hyperpolarization

The von Neumann entropy in thermal equilibrium at temperature $T$ is given by

$$S_{\text{vN}}^{\text{eq}}(T) = -\text{Tr}\{\rho_{\text{eq}}(T)\ln\rho_{\text{eq}}(T)\} \tag{38}$$

where the thermal equilibrium density operator is given by equation 36. This suggests the following criterion of hyperpolarization:

$$S_{\text{vN}} < S_{\text{vN}}^{\text{eq}}(T) \quad \text{(criterion of hyperpolarization)} \tag{39}$$

where $T$ is the temperature of the environment. Note that this definition of hyperpolarization makes no explicit mention of 290   population differences, or the existence of a net magnetic moment in a certain direction.

     The criterion in equation 39 identifies a region of Liouville space which is occupied by hyperpolarized states. For example, since the black dot in figure 7 indicates the thermal equilibrium density operator for spins $I = 1$ at temperature $T$, the contour line $S_{\text{vN}} = S_{\text{vN}}^{\text{eq}}(T)$ delineates the region of hyperpolarization at the temperature $T$. All density operators which are inside the grey region represent physically realisable hyperpolarized states of the spin ensemble.

Being inside the red region is a sufficient but not necessary criterion of hyperpolarization. Points outside the red region mights also represent hyperpolarized states, in the case that polarization moments which are not represented on the diagram, i.e. $\rho_{\lambda\mu}$ with $\{\lambda,\mu\} \neq \{1,0\}$ and $\{2,0\}$, are sufficiently large.

     The criterion in equation 39 is readily applied to higher spin systems, including coupled spin systems. For example, parahydrogen is hyperpolarized, since the corresponding density operator has a von Neumann entropy of zero, which is lower than 300   that of any thermal equilibrium state at finite temperature, even though pure parahydrogen possesses no magnetic moment or net angular momentum in a given direction.

## 7    Non-equilibrium spin dynamics

The dynamics of the spin density operator is governed by a differential equation called the *master equation* which takes into account coherent influences on the spin system (such as external magnetic fields and non-fluctuating components of the spin





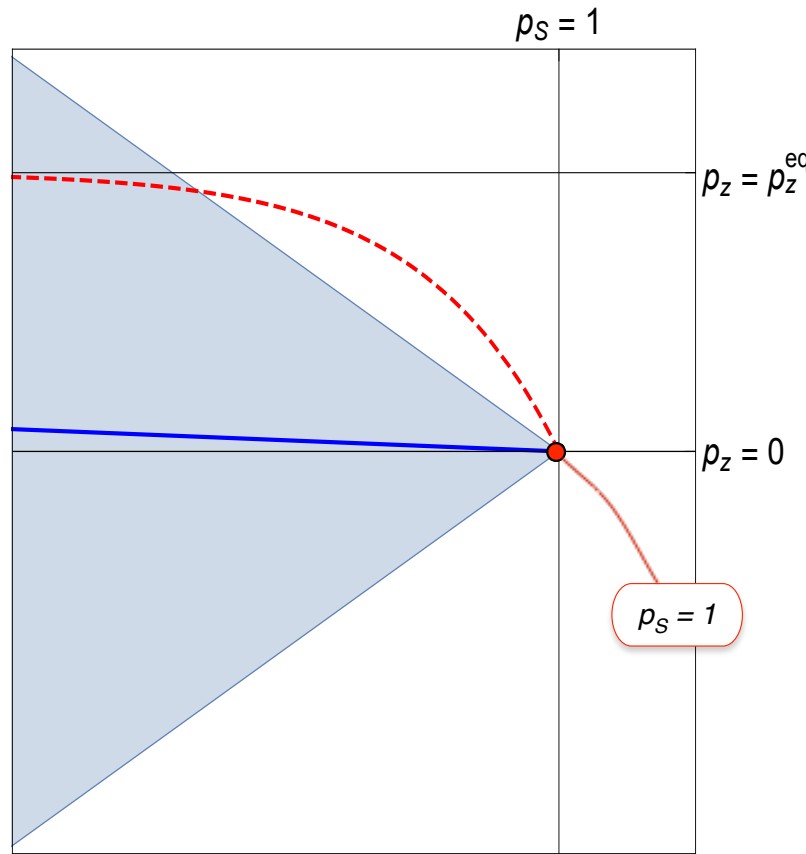

**Figure 8.** Non-equilibrium spin dynamics for an ensemble of spin-1/2 pairs, with $T_S \gg T_1$, where $T_S$ is the rate constant for the decay of singlet order, and $T_1$ is the rate constant for the equilibration of z-polarization. The plot shows an expanded view of figure 4 in the vicinity of the red dot, which represents an initial state of 100% singlet polarization. Dashed red line: Trajectory predicted by the inhomogeneous master equation; Solid blue line: Trajectory predicted by the Lindbladian master equation.

interactions) as well as relaxation effects. Various forms of the master equation have been proposed. The most widely used form is called the *inhomogeneous master equation*, which has the following form:

$$\frac{d}{dt}\rho = -i\hat{H}_{\mathrm{coh}}\rho(t) + \hat{\Gamma}\big(\rho(t) - \rho_{\mathrm{eq}}\big) \tag{40}$$

where $\hat{H}_{\mathrm{coh}}$ is the commutation superoperator of the coherent Hamiltonian, and $\hat{\Gamma}$ is the relaxation superoperator (Redfield (1965); Abragam (1961); Ernst et al. (1987)). The equilibrium spin density operator $\rho_{\mathrm{eq}}$ is given by equation 36.

The inhomogeneous master equation 40 is valid for high-entropy states which are close to equilibrium and is a standard component of NMR theory (Redfield (1965); Abragam (1961); Ernst et al. (1987)). However, in a previous paper (Bengs and Levitt (2020)), we showed that equation 40 loses validity for low-entropy states and may, in some cases, leads to non-physical predictions. We proposed a Lindbladian master equation, which has a wider range of validity.

**MAGNETIC RESONANCE**
Discussions
This point is reinforced by figure 8, which compares the predictions of the inhomogeneous and Lindbladian master equations when applied to spin-1/2 pairs in a low-entropy state of pure singlet polarization. The initial state of pure singlet order is shown by the red dot. The plot shows an expanded view of Liouville space, in the vicinity of the initial condition. The physical bounds of Liouville space are indicated by the blue triangle, as in figure 4.

The red dashed line shows the trajectory predicted by the IME, in the case that $T_S \gg T_1$, where $T_S$ is the relaxation time constant for singlet order (Levitt (2019)), and $T_1$ is the relaxation time constant for z-polarization. Since $T_1$ is relatively short, the z-polarization rapidly assumes its thermal equilibrium value $\rho_z^{\mathrm{eq}}$, which is finite in the presence of a strong magnetic field. However, as shown in figure 8, this leads the density operator into a forbidden region outside the physical boundary of Liouville space. This proves that the inhomogeneous master equation must be invalid in this regime.

The predicted trajectory of the Lindbladian master equation, as described in Bengs and Levitt (2020) is shown by the blue line. This uneventful trajectory always stays well within the physical boundary of Liouville space.

## 8 Conclusions

This article has been an exploration of the geometry and physical boundary of Liouville space, the home territory of all spin density operators. In the past, most NMR experiments have only explored a tiny region of this space, very close to the origin (except for the fixed projection onto the unity operator). However, NMR experiments are increasingly performed on highly non-equilibrium spin states, which are sometimes located on or near the physical Liouville space boundary. We hope that this article is useful as a guide for wanderers in this region.

### Acknowledgements

This paper is dedicated to the memory of Kostya Ivanov: our collaboration was fruitful and inspiring, but too brief. The research was supported by EPSRC(UK), grants EP/P009980/1, and the European Research Council, grant 786707-FunMagResBeacons. We also thank Jean-Nicolas Dumez, Thomas Schulte-Herbrüggen, Dima Budker, and Phil Kuchel for discussions. M.H.L. would like to express his heartfelt thanks to Geoffrey Bodenhausen for many years of friendship, guidance, support, encouragement, illuminating insights, and furious arguments.





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
