# Peer review of "Hyperpolarization and the Physical Boundary of Liouville Space"

_Magnetic Resonance, 2021_

## Community Comment (CC3)

This is a remarkable paper. All by itself, Figure 8 provides an elegant graphical demonstration of different predictions of Lindblad's and Redfield's formalisms. The paper further introduces a valuable interpretation of von Neumann's entropy, and defines what is meant by "hyperpolarization". Three papers for the price of one! Eq. (39) may well end up being recognized for Levitt like Schrödinger's equation has been for Schrödinger…

Figure 8 is a delight. Redfield, at least in the way his theory is commonly understood, predicts a curved trajectory that crosses the border into forbidden territory, while Lindblad predicts a straight "uneventful" trajectory that goes to the target along the shortest path. Euclid would have enjoyed this. Napoleon is supposed to have said: *un bon croquis vaut mieux qu'un long discours*. Apocryphal or not, the statement applies remarkably well to Figure 8.

Figure 8 reminds me of a property of Bloch's equations that is rarely discussed in the literature, although I always mentioned it in my undergraduate lectures. As long as pulses are used to drive the magnetization vector, its tip is confined to the surface of the sphere. If relaxation comes into play however, the interior of the sphere becomes accessible. In the extreme narrowing limit, the equations of motion of the transverse components and of the deviation of the longitudinal component from its equilibrium are the same. Hence, the trajectories follow a straight line, regardless where one starts on the surface. If one starts on the South pole, relaxation causes the vector to move towards the North pole along the z-axis, without ever creating any coherences, i.e., along the regular simplex of order one, or "1-simplex", as I discover reading Levitt and Bengs. Although Figure 8 is not concerned with coherences and describes a plane in a 4-dimensional space spanned by four eigenstates of two spins with $I = \frac{1}{2}$, the analogy between the two straight paths is striking.

Figure 8 brings new life to the debate between Lindblad vs Redfield. In discussions with clever colleagues, I have come to believe that the discrepancy stems not from the operator equations, but from truncations in the calculations. Redfield's double commutator is only the second term in a series expansion. What happens if one includes further terms? Would Redfield give the same predictions as Lindblad?

Figure 7 is another source of delight! Such curved trajectories neatly illustrate what happens in dissolution DNP.

Most of the paper by Levitt and Bengs explicitly excludes coherences, and focuses on eigenstates and their populations. It is a pity that the distinction between populations and zero-quantum coherences is not always clear. It is true that there are 6 operators of order $p = 0$ in a system with two spins $I = 1/2$, but they really should be broken up into 4 eigenstates with real populations and 1 zero-quantum coherence with a complex coefficient.

The skinny picture of the bare triangle of Figure 1 makes me feel nostalgic for Bloch's spherical bounty. How would Figure 1 be "fleshed out" if you incorporated coherences? I remember from the work of Karthik Gopalakrishnan that it can make sense to incorporate coherences that span states of different symmetry, e.g., between a singlet state and a triplet state. I originally resisted Karthik's very idea because I believed it to be paradoxical, but I was compelled to recognize the existence of long-lived coherences.

I agree with the anonymous referee no 2 that "polarization" is most clearly defined in NMR for a two-level system. In other areas of physics, the expression is usually associated with some vectorial (rank 1) quantity such as an electric dipole that results from a separation of charges. There are oodles of other meanings in physics and chemistry. Why should one argue about the semantic field of "polarization" in NMR?

I like the idea of "latent" or "hidden" (hyper)polarization suggested by referee no 2 to describe para-hydrogen experiments.

I do not understand why Levitt replaces his own brilliant invention of the expression "population imbalance" (that I have come to enjoy so much) by re-introducing the ambiguous term "singlet polarization". I agree with referee no 2 that "singlet polarization" is an oxymoron!

Personally, I like "longitudinal two spin order" or simply "zz order" more than "dipolar order".

There is nothing wrong with "(NISTOs)" for normalized irreducible spherical tensor operators. Personally, I much prefer the symbol $T\lambda\mu$ which says it all.

NMR people have become accustomed to "coefficients" to describe the weights of operators. Why bother to replace the popular "coefficients" by clumsy "polarization moments"? It is not because the expression has been used by atomic physicists like Auzinsh that it is necessarily more elegant than NMR jargon. After all, expression like "quantum error correction" are a paragon of clumsiness.

Some pedantic details: what is meant by "This is a region *exceedingly* close to the origin of Liouville space". Actually, the distance is on the order of 1 ppm in Boltzmann's equilibrium in high field at room temperature.

I do not understand the numbers along the scales of Fig. 2. How is the coefficient (or "polarization moment") of $T_{00}$ represented graphically?

The discussion of spin ½ pairs is valid only for magnetically *equivalent* pairs, is it not?

Despite my numerous above attempts to argue with the authors, I hope that my comments will be recognized for their illuminating *intention*, rather than taken to be furious arguments.

I note that this paper has stimulated a lively debate in the "discussions" of "Magnetic Resonance" in a manner that I have never experienced with any papers published in mainstream journals. Long live "MR"!

---

## Author Response (AR1)

We have made changes to the article in response to the referees' and community comments, as follows:

Referee #1

We have corrected the equation numbering.
We have replaced the term "zero-quantum" around line 75 by a more general statement (see highlighted version).

Referee #2
This referee, and some of the community commentators, raised semantic issues over our use of the term "polarization" in some contexts. Our view is that this term is more flexible than stated by the commentators and that its usage may be extended to cover our cases, for the sake of simplicity and economy of nomenclature. We have commented extensively on this throughout the revised manuscript (around lines 105 and 169).

- Could the author add more information on the condition in the eq. (23). Where exactly does it come from (i.e., why a sum of lambda from 1 to 2I)?
  - This has now been done (~ line 180)
- In Figure 4, -1/3 is listed as a lower bound but one can clearly see that the intersect with the x-axis (representing a contribution of the rank 0 polarization moment) is lower than |-0.3| (while it should cross at |-0.33|). Is it a representation error?
  - No, the referee has misunderstood the axes. We have clarified the scales used (caption to Figure 4).

- A similar analysis of the absence of information in the nuclear spin system was recently performed and could be mentioned with respect to the eq. (35): https://doi.org/10.1038/s41467-019-10787-9
  - We do not see the relevance of the article and have not made changes.
  -
- Page 17. "being inside the red region" – what exactly is the red region?
  - The references to red have been corrected to dark grey (lines 320-325).

- In Figure 7, "hyperpolarized states" are mentioned. This is clearly NMR jargon. States can be overpopulated or depleted but not hyperpolarized.
    - The states refereed to here belong to the spin ensemble, not to individual spin systems. The referee seems not to appreciate that the ensemble may have a state, described the density operator, and that the ensemble state may indeed be hyperpolarized. A comment has been added to the caption of figure 7, to clarify this.
    -
- I am not sure I fully understand Figure 8. Why does not polarization come to the equilibrium polarization eventually, even in the case of the Lindbladian equation? From the graph, it looks like it cannot ever come there..
    - The caption to figure 8 has been modified to make it clearer that both trajectories do lead to the same equilibrium state.